# Bayesian Prediction of Future Street Scenes using Synthetic Likelihoods.

**Apratim Bhattacharyya, Mario Fritz, Bernt Schiele**
Max Planck Institute for Informatics, Saarland Informatics Campus, Saarbrücken, Germany
{abhattac, mfritz, schiele}@mpi-inf.mpg.de

## Abstract

For autonomous agents to successfully operate in the real world, the ability to anticipate future scene states is a key competence. In real-world scenarios, future states become increasingly uncertain and multi-modal, particularly on long time horizons. Dropout based Bayesian inference provides a computationally tractable, theoretically well grounded approach to learn likely hypotheses/models to deal with uncertain futures and make predictions that correspond well to observations – are well calibrated. However, it turns out that such approaches fall short to capture complex real-world scenes, even falling behind in accuracy when compared to the plain deterministic approaches. This is because the used log-likelihood estimate discourages diversity. In this work, we propose a novel Bayesian formulation for anticipating future scene states which leverages synthetic likelihoods that encourage the learning of diverse models to accurately capture the multi-modal nature of future scene states. We show that our approach achieves accurate state-of-the-art predictions and calibrated probabilities through extensive experiments for scene anticipation on Cityscapes dataset. Moreover, we show that our approach generalizes across diverse tasks such as digit generation and precipitation forecasting.

## 1 Introduction

The ability to anticipate future scene states which involves mapping one scene state to likely future states under uncertainty is key for autonomous agents to successfully operate in the real world e.g., to anticipate the movements of pedestrians and vehicles for autonomous vehicles. The future states of street scenes are inherently uncertain and the distribution of outcomes is often multi-modal. This is especially true for important classes like pedestrians. Recent works on anticipating street scenes (Luc et al., 2017; Jin et al., 2017; Seyed et al., 2018) do not systematically consider uncertainty.

Bayesian inference provides a theoretically well founded approach to capture both model and observation uncertainty but with considerable computational overhead. A recently proposed approach (Gal & Ghahramani, 2016b; Kendall & Gal, 2017) uses dropout to represent the posterior distribution of models and capture model uncertainty. This approach has enabled Bayesian inference with deep neural networks without additional computational overhead. Moreover, it allows the use of any existing deep neural network architecture with minor changes.

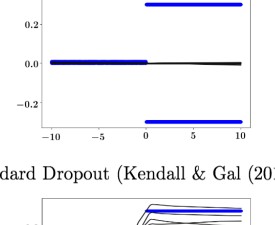

Standard Dropout (Kendall & Gal (2017))

With Our Synthetic Likelihoods

**Figure 1:** Blue: Groundtruth distribution. Black: Models sampled at random from the model distribution.

However, when the underlying data distribution is multimodal and the model set under consideration do not have explicit latent state/variables (as most popular deep deep neural network architectures), the approach of Gal & Ghahramani (2016b); Kendall & Gal (2017) is unable to recover the true model uncertainty (see Figure 1 and Osband (2016)). This is because this approach is known to conflate risk and uncertainty (Osband, 2016). This limits the accuracy of the models over a plain deterministic (non-Bayesian) approach. The main cause is the data log-likelihood maximization step

during optimization – for every data point the average likelihood assigned by all models is maximized. This forces every model to explain every data point well, pushing every model in the distribution to the mean. We address this problem through an objective leveraging synthetic likelihoods (Wood, 2010; Rosca et al., 2017) which relaxes the constraint on every model to explain every data point, thus encouraging diversity in the learned models to deal with multi-modality.

In this work: 1. We develop the first Bayesian approach to anticipate the multi-modal future of street scenes and demonstrate state-of-the-art accuracy on the diverse Cityscapes dataset without compromising on calibrated probabilities, 2. We propose a novel optimization scheme for dropout based Bayesian inference using synthetic likelihoods to encourage diversity and accurately capture model uncertainty, 3. Finally, we show that our approach is not limited to street scenes and generalizes across diverse tasks such as digit generation and precipitation forecasting.

## 2 RELATED WORK

**Bayesian deep learning.** Most popular deep learning models do not model uncertainty, only a mean model is learned. Bayesian methods (MacKay, 1992; Neal, 2012) on the other hand learn the posterior distribution of likely models. However, inference of the model posterior is computationally expensive. In (Gal & Ghahramani, 2016b) this problem is tackled using variational inference with an approximate Bernoulli distribution on the weights and the equivalence to dropout training is shown. This method is further extended to convolutional neural networks in (Gal & Ghahramani, 2016a). In (Kendall & Gal, 2017) this method is extended to tackle both model and observation uncertainty through heteroscedastic regression. The proposed method achieves state of the art results on segmentation estimation and depth regression tasks. This framework is used in Bhattacharyya et al. (2018a) to estimate future pedestrian trajectories. In contrast, Saatci & Wilson (2017) propose a (unconditional) Bayesian GAN framework for image generation using Hamiltonian Monte-Carlo based optimization with limited success. Moreover, conditional variants of GANs (Mirza & Osindero, 2014) are known to be especially prone to mode collapse. Therefore, we choose a dropout based Bayesian scheme and improve upon it through the use of synthetic likelihoods to tackle the issues with model uncertainty mentioned in the introduction.

**Structured output prediction.** Stochastic feedforward neural networks (SFNN) and conditional variational autoencoders (CVAE) have also shown success in modeling multimodal conditional distributions. SFNNs are difficult to optimize on large datasets (Tang & Salakhutdinov, 2013) due to the binary stochastic variables. Although there has been significant effort in improving training efficiency (Rezende et al., 2014; Gu et al., 2016), success has been partial. In contrast, CVAEs (Sohn et al., 2015) assume Gaussian stochastic variables, which are easier to optimize on large datasets using the re-parameterization trick. CVAEs have been successfully applied on a large variety of tasks, include conditional image generation (Bao et al., 2017), next frame synthesis (Xue et al., 2016), video generation (Babaeizadeh et al., 2018; Denton & Fergus, 2018), trajectory prediction (Lee et al., 2017) among others. The basic CVAE framework is improved upon in (Bhattacharyya et al., 2018b) through the use of a multiple-sample objective. However, in comparison to Bayesian methods, careful architecture selection is required and experimental evidence of uncertainty calibration is missing. Calibrated uncertainties are important for autonomous/assisted driving, as users need to be able to express trust in the predictions for effective decision making. Therefore, we also adopt a Bayesian approach over SFNN or CVAE approaches.

**Anticipation future scene scenes.** In (Luc et al., 2017) the first method for predicting future scene segmentations has been proposed. Their model is fully convolutional with prediction at multiple scales and is trained auto-regressively. Jin et al. (2017) improves upon this through the joint prediction of future scene segmentation and optical flow. Similar to Luc et al. (2017) a fully convolutional model is proposed, but the proposed model is based on the Resnet-101 (He et al., 2016) and has a single prediction scale. More recently, Luc et al. (2018) has extended the model of Luc et al. (2017) to the related task of future instance segmentation prediction. These methods achieve promising results and establish the competence of fully convolutional models. In (Seyed et al., 2018) a Convolutional LSTM based model is proposed, further improving short-term results over Jin et al. (2017). However, fully convolutional architectures have performed well at a variety of related tasks, including segmentation estimation (Yu & Koltun, 2016; Zhao et al., 2017), RGB frame prediction

(Mathieu et al., 2016; Babaeizadeh et al., 2018) among others. Therefore, we adopt a standard ResNet based fully-convolutional architecture, while providing a full Bayesian treatment.

## 3 BAYESIAN MODELS FOR PREDICTION UNDER UNCERTAINTY

We phrase our models in a Bayesian framework, to jointly capture model (epistemic) and observation (aleatoric) uncertainty (Kendall & Gal, 2017). We begin with model uncertainty.

### 3.1 MODEL UNCERTAINTY

Let $x \in X$ be the input (past) and $y \in Y$ be the corresponding outcomes. Consider $f : x \mapsto y$, we capture model uncertainty by learning the distribution $p(f|X, Y)$ of generative models $f$, likely to have generated our data $\{X, Y\}$. The complete predictive distribution of outcomes $y$ is obtained by marginalizing over the posterior distribution,

$$p(y|x, X, Y) = \int p(y|x, f)p(f|X, Y)df .$$ (1)

However, the integral in (1) is intractable. But, we can approximate it in two steps (Gal & Ghahramani, 2016b). First, we assume that our models can be described by a finite set of variables $\omega$. Thus, we constrain the set of possible models to ones that can be described with $\omega$. Now, (1) is equivalently,

$$p(y|x, X, Y) = \int p(y|x, \omega)p(\omega|X, Y)d\omega .$$ (2)

Second, we assume an approximating variational distribution $q(\omega)$ of models which allows for efficient sampling. This results in the approximate distribution,

$$p(y|x, X, Y) \approx p(y|x) = \int p(y|x, \omega)q(\omega)d\omega .$$ (3)

For convolutional models, Gal & Ghahramani (2016a) proposed a Bernoulli variational distribution defined over each convolutional patch. The number of possible models is exponential in the number of patches. This number could be very large, making it difficult optimize over this very large set of models. In contrast, in our approach (4), the number possible models is exponential in the number of weight parameters, a much smaller number. In detail, we choose the set of convolutional kernels and the biases $\{(W_1, b_1), \ldots, (W_L, b_L)\} \in \mathcal{W}$ of our model as the set of variables $\omega$. Then, we define the following novel approximating Bernoulli variational distribution $q(\omega)$ independently over each element $w_{k',k}^{i,j}$ (correspondingly $b_k$) of the kernels and the biases at spatial locations $\{i, j\}$,

$$q(W_K) = M_K \odot Z_K$$
$$z_{k',k}^{i,j} = \text{Bernoulli}(p_K), \quad k' = 1, \ldots, |K'|, \quad k = 1, \ldots, |K| .$$ (4)

Note, $\odot$ denotes the hadamard product, $M_k$ are tuneable variational parameters, $z_{k',k}^{i,j} \in Z_K$ are the independent Bernoulli variables, $p_K$ is a probability tensor equal to the size of the (bias) layer, $|K|$ ($|K'|$) is the number of kernels in the current (previous) layer. Here, $p_K$ is chosen manually. Moreover, in contrast to Gal & Ghahramani (2016a), the same (sampled) kernel is applied at each spatial location leading to the detection of the same features at varying spatial locations. Next, we describe how we capture observation uncertainty.

### 3.2 OBSERVATION UNCERTAINTY

Observation uncertainty can be captured by assuming an appropriate distribution of observation noise and predicting the sufficient statistics of the distribution (Kendall & Gal, 2017). Here, we assume a Gaussian distribution with diagonal covariance matrix at each pixel and predict the mean vector $\mu^{i,j}$ and co-variance matrix $\sigma^{i,j}$ of the distribution. In detail, the predictive distribution of a generative model draw from $\hat{\omega} \sim q(\omega)$ at a pixel position $\{i, j\}$ is,

$$p^{i,j}(y|x, \hat{\omega}) = \mathcal{N}\big((\mu^{i,j}|x, \hat{\omega}), (\sigma^{i,j}|x, \hat{\omega})\big) .$$ (5)

We can sample from the predictive distribution $p(\mathbf{y}|\mathbf{x})$ (3) by first sampling the weight matrices $\omega$ from (4) and then sampling from the Gaussian distribution in (5). We perform the last step by the linear transformation of a zero mean unit diagonal variance Gaussian, ensuring differentiability,

$$\hat{\mathbf{y}}^{i,j} \sim \mu^{i,j}(\mathbf{x}|\hat{\omega}) + z \times \sigma^{i,j}(\mathbf{x}|\hat{\omega}), \quad \text{where } p(z) \text{ is } \mathcal{N}(0, I) \text{ and } \hat{\omega} \sim q(\omega). \tag{6}$$

where, $\hat{\mathbf{y}}^{i,j}$ is the sample drawn at a pixel position $\{i, j\}$ through the liner transformation of $z$ (a vector) with the predicted mean $\mu^{i,j}$ and variance $\sigma^{i,j}$. In case of street scenes, $\mathbf{y}^{i,j}$ is a class-confidence vector and sample of final class probabilities is obtained by pushing $\hat{\mathbf{y}}^{i,j}$ through a softmax.

## 3.3 TRAINING

For a good variational approximation (3), our approximating variational distribution of generative models $q(\omega)$ should be close to the true posterior $p(\omega|\mathbf{X}, \mathbf{Y})$. Therefore, we minimize the KL divergence between these two distributions. As shown in Gal & Ghahramani (2016b;a); Kendall & Gal (2017) the KL divergence is given by (over i.i.d data points),

$$\mathrm{KL}(q(\omega) \,||\, p(\omega|\mathbf{X}, \mathbf{Y})) \propto \mathrm{KL}(q(\omega) \,||\, p(\omega)) - \int q(\omega) \log p(\mathbf{Y}|\mathbf{X}, \omega) d\omega$$

$$= \mathrm{KL}(q(\omega) \,||\, p(\omega)) - \int q(\omega) \Big( \int \log p(\mathbf{y}|\mathbf{x}, \omega) d(\mathbf{x}, \mathbf{y}) \Big) d\omega. \tag{7}$$

$$= \mathrm{KL}(q(\omega) \,||\, p(\omega)) - \int \Big( \int q(\omega) \log p(\mathbf{y}|\mathbf{x}, \omega) d\omega \Big) d(\mathbf{x}, \mathbf{y}).$$

The log-likelihood term at the right of (7) considers every model for every data point. This imposes the constraint that every data point must be explained well by every model. However, if the data distribution $(\mathbf{x}, \mathbf{y})$ is multi-modal, this would push every model to the mean of the multi-modal distribution (as in Figure 1 where only way for models to explain both modes is to converge to the mean). This discourages diversity in the learned modes. In case of multi-modal data, we would not be able to recover all likely models, thus hindering our ability to fully capture model uncertainty. The models would be forced to explain the data variation as observation noise (Osband, 2016), thus conflating model and observation uncertainty. We propose to mitigate this problem through the use of an approximate objective using synthetic likelihoods (Wood, 2010; Rosca et al., 2017) – obtained from a classifier. The classifier estimates the likelihood based on whether the models $\hat{\omega} \sim q(\omega)$ explain (generate) data samples likely under the true data distribution $p(\mathbf{y}|\mathbf{x})$. This removes the constraint on models to explain every data point – it only requires the explained (generated) data points to be likely under the data distribution. Thus, this allows models $\hat{\omega} \sim q(\omega)$ to be diverse and deal with multi-modality. Next, we reformulate the KL divergence estimate of (7) to a likelihood ratio form which allows us to use a classifier to estimate (synthetic) likelihoods, (also see Appendix),

$$= \mathrm{KL}(q(\omega) \,||\, p(\omega)) - \int \Big( \int q(\omega) \log p(\mathbf{y}|\mathbf{x}, \omega) d\omega \Big) d(\mathbf{x}, \mathbf{y}).$$

$$= \mathrm{KL}(q(\omega) \,||\, p(\omega)) - \int \Big( \int q(\omega) \big( \log \frac{p(\mathbf{y}|\mathbf{x}, \omega)}{p(\mathbf{y}|\mathbf{x})} + \log p(\mathbf{y}|\mathbf{x}) \big) d\omega \Big) d(\mathbf{x}, \mathbf{y}). \tag{8}$$

$$\propto \mathrm{KL}(q(\omega) \,||\, p(\omega)) - \int \int q(\omega) \log \frac{p(\mathbf{y}|\mathbf{x}, \omega)}{p(\mathbf{y}|\mathbf{x})} d\omega \; d(\mathbf{x}, \mathbf{y}).$$

In the second step of (8), we divide and multiply the probability assigned to a data sample by a model $p(\mathbf{y}|\mathbf{x}, \omega)$ by the true conditional probability $p(\mathbf{y}|\mathbf{x})$ to obtain a likelihood ratio. We can estimate the KL divergence by equivalently estimating this ratio rather than the true likelihood. In order to (synthetically) estimate this likelihood ratio, let us introduce the variable $\theta$ to denote, $p(\mathbf{y}|\mathbf{x}, \theta = 1)$ the probability assigned by our model $\omega$ to a data sample $(\mathbf{x}, \mathbf{y})$ and $p(\mathbf{y}|\mathbf{x}, \theta = 0)$ the true probability of the sample. Therefore, the ratio in the last term of (8) is,

$$= \mathrm{KL}(q(\omega) \,||\, p(\omega)) - \int \int q(\omega) \log \frac{p(\mathbf{y}|\mathbf{x}, \theta = 1)}{p(\mathbf{y}|\mathbf{x}, \theta = 0)} d\omega \; d(\mathbf{x}, \mathbf{y}).$$

$$= \mathrm{KL}(q(\omega) \,||\, p(\omega)) - \int \int q(\omega) \log \frac{p(\theta = 1|\mathbf{x}, \mathbf{y})}{p(\theta = 0|\mathbf{x}, \mathbf{y})} d\omega \; d(\mathbf{x}, \mathbf{y}). \quad \text{(Using Bayes theorem)} \tag{9}$$

$$= \mathrm{KL}(q(\omega) \,||\, p(\omega)) - \int \int q(\omega) \log \frac{p(\theta = 1|\mathbf{x}, \mathbf{y})}{1 - p(\theta = 1|\mathbf{x}, \mathbf{y})} d\omega \; d(\mathbf{x}, \mathbf{y}).$$

In the last step of (9) we use the fact that the events $\theta = 1$ and $\theta = 0$ are mutually exclusive. We can approximate the ratio $\frac{p(\theta=1|\mathrm{x},\mathrm{y})}{1-p(\theta=1|\mathrm{x},\mathrm{y})}$ by jointly learning a discriminator $D(\mathrm{x}, \hat{\mathrm{y}})$ that can distinguish between samples of the true data distribution and samples $(\mathrm{x}, \hat{\mathrm{y}})$ generated by the model $\omega$, which provides a synthetic estimate of the likelihood, and equivalently integrating directly over $(\mathrm{x}, \hat{\mathrm{y}})$,

$$\approx \mathrm{KL}(q(\omega) \mid\mid p(\omega)) - \int \int q(\omega) \log \Big( \frac{D(\mathrm{x}, \hat{\mathrm{y}})}{1 - D(\mathrm{x}, \hat{\mathrm{y}})} \Big) d\omega \; d(\mathrm{x}, \hat{\mathrm{y}}). \tag{10}$$

Note that the synthetic likelihood $\big( \frac{D(\mathrm{x},\hat{\mathrm{y}})}{1-D(\mathrm{x},\hat{\mathrm{y}})} \big)$ is independent of any specific pair $(\mathrm{x}, \mathrm{y})$ of the true data distribution (unlike the log-likelihood term in (7)), its value depends only upon whether the generated data point $(\mathrm{x}, \hat{\mathrm{y}})$ by the model $\omega$ is likely under the true data distribution $p(\mathrm{y}|\mathrm{x})$. Therefore, the models $\omega$ have to only generate samples $(\mathrm{x}, \hat{\mathrm{y}})$ likely under the true data distribution. The models need not explain every data point equally well. Therefore, we do not push the models $\omega$ to the mean, thus allowing them to be diverse and allowing us to better capture uncertainty.

Empirically, we observe that a hybrid log-likelihood term using both the log-likelihood terms of (10) and (7) with regularization parameters $\alpha$ and $\beta$ (with $\alpha \geq \beta$) stabilizes the training process,

$$\alpha \int \int q(\omega) \log \Big( \frac{D(\mathrm{x}, \hat{\mathrm{y}})}{1 - D(\mathrm{x}, \hat{\mathrm{y}})} \Big) d\omega \; d(\mathrm{x}, \hat{\mathrm{y}}) + \beta \int \int q(\omega) \log p(\mathrm{y}|\mathrm{x}, \omega) d\omega \; d(\mathrm{x}, \mathrm{y}). \tag{11}$$

Note that, although we do not explicitly require the posterior model distribution to explain all data points, due to the exponential number of models afforded by dropout and the joint optimization (min-max game) of the discriminator, empirically we see very diverse models explaining most data points. Moreover, empirically we also see that predicted probabilities remain calibrated. Next, we describe the architecture details of our generative models $\omega$ and the discriminator $D(\mathrm{x}, \hat{\mathrm{y}})$.

## 3.4 Model architecture for street scene prediction

The architecture of our ResNet based generative models in our model distribution $q(\omega)$ is shown in Figure 2. The generative model takes as input a sequence of past segmentation class-confidences $\mathrm{s_p}$, the past and future vehicle odometry $\mathrm{o_p}$, $\mathrm{o_f}$ ($\mathrm{x} = \{\mathrm{s_p}, \mathrm{o_p}, \mathrm{o_f}\}$) and produces the class-confidences at the next time-step as output. The additional conditioning on vehicle odometry is because the sequences are recorded in frame of reference of a moving vehicle and therefore the future observed sequence is dependent upon the vehicle trajectory. We use recursion to efficiently predict a sequence of future scene segmentations $\mathrm{y} = \{\mathrm{s_f}\}$. The discriminator takes as input $\mathrm{s_f}$ and classifies whether it was produced by our model or is from the true data distribution.

In detail, generative model architecture consists of a fully convolutional encoder-decoder pair. This architecture builds upon prior work of Luc et al. (2017); Jin et al. (2017), however with key differences. In Luc et al. (2017), each of the two levels of the model architecture consists of only five convolutional layers. In contrast, our model consists of one level with five convolutaional blocks. The encoder contains three residual blocks with max-pooling in between and the decoder consists of a residual and a convoluational block with up-sampling in between. We double the size of the blocks following max-pooling in

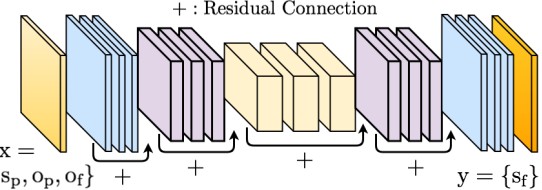

**Figure 2:** The architecture of our ResNet based generative models for street scene prediction in our model distribution $q(\omega)$.

order to preserve resolution. This leads to a much deeper model with fifteen convolutional layers, with constant spatial convolutional kernel sizes. This deep model with pooling creates a wide receptive field and helps better capture spatio-temporal dependencies. The residual connections help in the optimization of such a deep model. Computational resources allowing, it is possible to add more levels to our model. In Jin et al. (2017) a model is considered which uses a Res101-FCN as an encoder. Although this model has significantly more layers, it also introduces a large amount of pooling. This leads to loss of resolution and spatial information, hence degrading performance. Our discriminator model consists of six convolutional layers with max-pooling layers in-between, followed by two fully connected layers. Finally, in Appendix E we provide layer-wise details and discuss the reduction of number of models in $q(\omega)$ through the use of Weight Dropout (4) for our architecture of generators.

| Groundtruth | Bayes-S Samples | Bayes-SL Samples | | Method | Top-10% |
|---|---|---|---|---|---|
| | | | | Mean | 80.1±0.4 |
| | | | | Bayes-S | 79.3±0.4 |
| | | | | CVAE | 82.5±0.5 |
| | | | | **Bayes-SL** | **83.1±0.6** |

**Figure 3: Left**: MNIST generations: The models see the non grayed-out region of the digit. The samples are generated from models drawn at random from $\hat{\omega} \sim q(\omega)$. **Right**: Top-10% accuracy on MNIST generation.

## 4 EXPERIMENTS

Next, we evaluate our approach on MNIST digit generation and street scene anticipation on Cityscapes. We further evaluate our model on 2D data (Figure 1) and precipitation forecasting in the Appendix.

### 4.1 MNIST DIGIT GENERATION

Here, we aim to generate the full MNIST digit given only the lower left quarter of the digit. This task serves as an ideal starting point as in many cases there are multiple likely completions given the lower left quarter digit, e.g. 5 and 3. Therefore, the learned model distribution $q(\omega)$ should contain likely models corresponding to these completions. We use a fully connected generator with 6000-4000-2000 hidden units with 50% dropout probability. The discriminator has 1000-1000 hidden units with leaky ReLU non-linearities. We set $\beta = 10^{-4}$ for the first 4 epochs and then reduce it to 0, to provide stability during the initial epochs. We compare our synthetic likelihood based approach (Bayes-SL) with, 1. A non-Bayesian mean model, 2. A standard Bayesian approach (Bayes-S), 3. A Conditional Variational Autoencoder (CVAE) (architecture as in Sohn et al. (2015)). As evaluation metric we consider (oracle) Top-k% accuracy (Lee et al., 2017). We use a standard Alex-Net based classifier to measure if the best prediction corresponds to the ground-truth class – identifies the correct mode – in Table 3 (right) over 10 splits of the MNIST test-set. We sample 10 models from our learned distribution and consider the best model. We see that our Bayes-SL performs best, even outperforming the CVAE model. In the qualitative examples in Table 3 (left), we see that generations from models $\hat{\omega} \sim q(\omega)$ sampled from our learned model distribution corresponds to clearly defined digits (also in comparision to Figure 3 in Sohn et al. (2015)). In contrast, we see that the Bayes-S model produces blurry digits. All sampled models have been pushed to the mean and shows little advantage over a mean model.

### 4.2 CITYSCAPES STREET SCENE ANTICIPATION

Next, we evaluate our apporach on the Cityscapes dataset – anticipating scenes more than 0.5 seconds into the future. The street scenes already display considerable multi-modality at this time-horizon.

**Evaluation metrics and baselines.** We use PSPNet Zhao et al. (2017) to segment the full training sequences as only the 20[th] frame has groundtruth annotations. We always use the annotated 20[th] frame of the validation sequences for evaluation using the standard mean Intersection-over-Union (mIoU) and the per-pixel (negative) conditional log-likelihood (CLL) metrics. We consider the following baselines for comparison to our Resnet based (architecture in Figure 2) Bayesian (Bayes-WD-SL) model with weight dropout and trained using synthetic likelihoods: 1. Copying the last seen input; 2. A non-Bayesian (ResG-Mean) version; 3. A Bayesian version with standard patch dropout (Bayes-S); 4. A Bayesian version with our weight dropout (Bayes-WD). Note that, combination of ResG-Mean with an adversarial loss did not lead to improved results (similar observations made in Luc et al. (2017)). We use grid search to set the dropout rate (in (4)) to 0.15 for the Bayes-S and 0.20 for Bayes-WD(-SL) models. We set $\alpha, \beta = 1$ for our Bayes-WD-SL model. We train all models using Adam (Kingma & Ba, 2015) for 50 epochs with batch size 8. We use one sample to train the Bayesian methods as in Gal & Ghahramani (2016a) and use 100 samples during evaluation.

**Comparison to state of the art.** We begin by comparing our Bayesian models to state-of-the-art methods Luc et al. (2017); Seyed et al. (2018) in Table 1. We use the mIoU metric and for a fair comparison consider the mean (of all samples) prediction of our Bayesian models. We alwyas compare to the groundtruth segmentations of the validation set. However, as all three methods use a slightly different semantic segmentation algorithm (Table 2) to generate training and input test data, we include the mIoU achieved by the Last Input of all three methods (see Appendix C for results

**Table 1:** Comparing mean predictions to the state-of-the-art.

| Method | Timestep | | |
| --- | --- | --- | --- |
| | +0.06sec | +0.18sec | +0.54sec |
| Last Input (Luc et al. (2017)) | x | 49.4 | 36.9 |
| Luc et al. (2017) (ft) | x | 59.4 | 47.8 |
| Last Input (Seyed et al. (2018)) | 62.6 | 51.0 | x |
| Seyed et al. (2018) | 71.3 | 60.0 | x |
| Last Input (Ours) | 67.1 | 52.1 | 38.3 |
| Bayes-S (mean) | 71.2 | 64.8 | 45.7 |
| Bayes-WD (mean) | 73.7 | 63.5 | 44.0 |
| Bayes-WD-SL (mean) | **74.1** | 64.8 | 45.9 |
| Bayes-WD-SL (ft, mean) | x | **65.1** | **51.2** |
| Bayes-WD-SL (top 5%) | **75.3** | 65.2 | 49.5 |
| Bayes-WD-SL (ft, top 5%) | x | **66.7** | **52.5** |

**Table 2:** Comparison of segmentation estimation methods on Cityscapes validation set.

| Method | mIoU |
| --- | --- |
| Dilation10 (Luc et al., 2017) | 68.8 |
| PSPNet (Seyed et al., 2018) | 75.7 |
| PSPNet (Ours) | 76.9 |

**Table 3:** Evaluation on capturing uncertainty (using mIoU top 5%).

| Method | Timestep | | | |
| --- | --- | --- | --- | --- |
| | $t+5$ | | $t+10$ | |
| | mIoU | CLL | mIoU | CLL |
| Last Input | 45.7 | 0.86 | 37.1 | 1.35 |
| ResG-Mean | 59.1 | 0.49 | 46.6 | 0.89 |
| Bayes-S | 58.8 | 0.48 | 46.1 | 0.80 |
| Bayes-WD | 59.2 | 0.48 | 46.6 | **0.79** |
| Bayes-WD-SL | **60.2** | **0.47** | **47.1** | **0.79** |

**Table 4:** Ablation study and comparison to a CVAE baseline.

| Method | Timestep | |
| --- | --- | --- |
| | $t+5$ | $t+10$ |
| | mIoU | mIoU |
| CVAE (First) | 58.7 | 45.5 |
| CVAE (Mid) | 58.9 | 46.6 |
| CVAE (Last) | 59.2 | 46.8 |
| Bayes-WD-SL | **60.2** | **47.1** |

using Dialation 10). Similar to Luc et al. (2017) we fine-tune (ft) to predict at 3 frame intervals for better performance at +0.54sec. Our Bayes-WD-SL model outperforms baselines and improves on prior work by 2.8 mIoU at +0.06sec and 4.8 mIoU/3.4 mIoU at +0.18sec/+0.54sec respectively. Our Bayes-WD-SL model also obtains higher relative gains in comparison to Luc et al. (2017) with respect the Last Input Baseline. These results validate our choice of model architecture and show that our novel approach clearly outperforms the state-of-the-art. The performance advantage of Bayes-WD-SL over Bayes-S shows that the ability to better model uncertainty does not come at the cost of lower mean performance. However, at larger time-steps as the future becomes increasingly uncertain, mean predictions (mean of all likely futures) drift further from the ground-truth. Therefore, next we evaluate the models on their (more important) ability to capture the uncertainty of the future.

**Evaluation of predicted uncertainty.** Next, we evaluate whether our Bayesian models are able to accurately capture uncertainty and deal with multi-modal futures, upto $t+10$ frames (0.6 seconds) in Table 3. We consider the mean of (oracle) best 5% of predictions (Lee et al. (2017)) of our Bayesian models to evaluate whether the learned model distribution $q(\omega)$ contains likely models corresponding to the groundtruth. We see that the best predictions considerably improve over the mean predictions – showing that our Bayesian models learns to capture uncertainity and deal with multi-modal futures. Quantitatively, we see that the Bayes-S model performs worst, demonstrating again that standard dropout (Kendall & Gal, 2017) struggles to recover the true model uncertainty. The use of weight dropout improves the performance to the level of the ResG-Mean model. Finally, we see that our Bayes-WD-SL model performs best. In fact, it is the only Bayesian model whose (best) performance exceeds that of the ResG-Mean model (also outperforming state-of-the-art), demonstrating the effectiveness of synthetic likelihoods during training. In Figure 5 we show examples comparing the best prediction of our Bayes-WD-SL model and ResG-Mean at $t+9$. The last row highlights the differences between the predictions – cyan shows areas where our Bayes-WD-SL is correct and ResG-Mean is wrong, red shows the opposite. We see that our Bayes-WD-SL performs better at classes like cars and pedestrians which are harder to predict (also in comparison to Table 5 in Luc et al. (2017)). In Figure 6, we show samples from randomly sampled models $\hat{\omega} \sim q(\omega)$, which shows correspondence to the range of possible movements of bicyclists/pedestrians. Next, we further evaluate the models with the CLL metric in Table 3. We consider the mean predictive distributions (3) up to $t+10$ frames. We see that the Bayesian models outperform the ResG-Mean model significantly. In particular, we see that our Bayes-WD-SL model performs the best, demonstrating that the learned model and observation uncertainty corresponds to the variation in the data.

**Comparison to a CVAE baseline.** As there exists no CVAE (Sohn et al., 2015) based model for future segmentation prediction, we construct a baseline as close as possible to our Bayesian models

| Groundtruth, $t + 9$ | ResG-Mean, $t + 9$ | Bayes-WD-SL, $t + 9$ | Comparison |
|---|---|---|---|

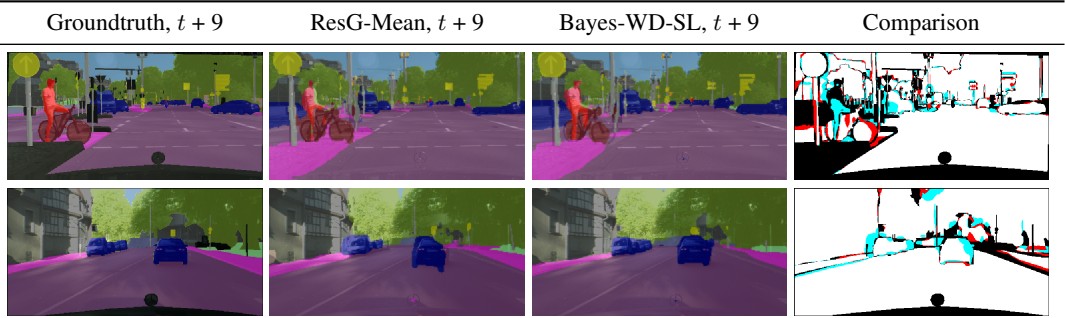

**Figure 5:** Bayes-WD-SL (top 1) vs ResG-Mean. Cyan: Bayes-WD-SL is correct and ResG-Mean is wrong. Red: Bayes-WD-SL is wrong and ResG-Mean is correct, white: both right, black: both wrong/unlabeled.

| Sample #1, $t + 9$ | Sample #2, $t + 9$ | Sample #3, $t + 9$ | Sample #4, $t + 9$ |
|---|---|---|---|

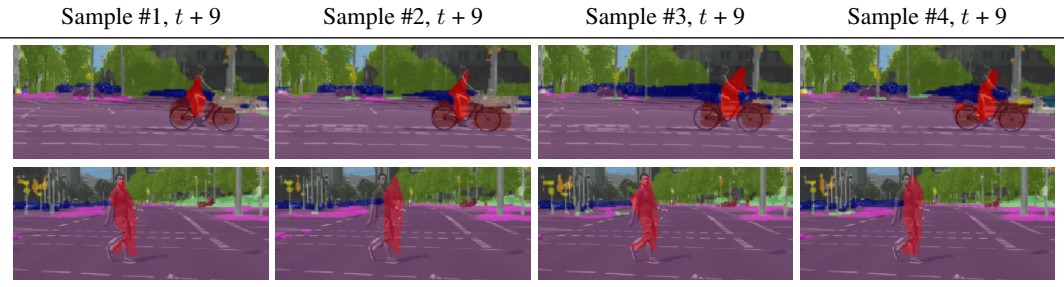

**Figure 6:** Random samples from our Bayes-WD-SL model corresponds to the range of likely movements of bicyclists/pedestrians.

based on existing CVAE based models for related tasks (Babaeizadeh et al., 2018; Xue et al., 2016). Existing CVAE based models (Babaeizadeh et al., 2018; Xue et al., 2016) contain a few layers with Gaussian input noise. Therefore, for a fair comparison we first conduct a study in Table 4 to find the layers which are most effective at capturing data variation. We consider Gaussian input noise applied in the first, middle or last convolutional blocks. The noise is input dependent during training, sampled from a recognition network (see Appendix). We observe that noise in the last layers can better capture data variation. This is because the last layers capture semantically higher level scene features. Overall, our Bayesian approach (Bayes-WD-SL) performs the best. This shows that the CVAE model is not able to effectively leverage Gaussian noise to match the data variation.

**Uncertainty calibration.** We further evaluate predicted uncertainties by measuring their calibration – the correspondence between the predicted probability of a class and the frequency of its occurrence in the data. As in Kendall & Gal (2017), we discretize the output probabilities of the mean predicted distribution into bins and measure the frequency of correct predictions for each bin. We report the results at $t + 10$ frames in Figure 4. We observe that all Bayesian approaches outperform the ResG-Mean and CVAE versions. This again demonstrates the effectiveness of the Bayesian approaches in capturing uncertainty.

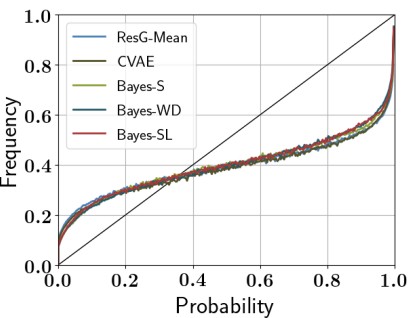

**Figure 4:** Uncertainty calibration at $t + 10$.

## 5    CONCLUSION

We propose a novel approach for predicting real-world semantic segmentations into the future that casts a convolutional deep learning approach into a Bayesian formulation. One of the key contributions is a novel optimization scheme that uses synthetic likelihoods to encourage diversity and deal with multi-modal futures. Our proposed method shows state of the art performance in challenging street scenes. More importantly, we show that the probabilistic output of our deep learning architecture captures uncertainty and multi-modality inherent to this task. Furthermore, we show that the developed methodology goes beyond just street scene anticipation and creates new opportunities to enhance high performance deep learning architectures with principled formulations of Bayesian inference.

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

APPENDIX A. DETAILED DERIVATIONS.

**KL divergence estimate.** Here, we provide a detailed derivation of (8). Starting from (7), we have,

$$
\begin{aligned}
&\mathrm{KL}(q(\omega|\mathbf{X},\mathbf{Y}) \,||\, p(\omega|\mathbf{X},\mathbf{Y})) \\
&\propto \mathrm{KL}(q(\omega) \,||\, p(\omega)) - \int q(\omega) \log p(\mathbf{Y}|\mathbf{X},\omega) d\omega. \\
&= \mathrm{KL}(q(\omega) \,||\, p(\omega)) - \int q(\omega) \Big( \int \log p(\mathbf{y}|\mathbf{x},\omega\,) d(\mathbf{x},\mathbf{y}) \Big) d\omega. \quad \text{(over i.i.d } (\mathbf{x},\mathbf{y}) \in (\mathbf{X},\mathbf{Y})) \\
&= \mathrm{KL}(q(\omega) \,||\, p(\omega)) - \int \Big( \int q(\omega) \log p(\mathbf{y}|\mathbf{x},\omega\,) d\omega \Big) d(\mathbf{x},\mathbf{y}).
\end{aligned}
\tag{S1}
$$

Multiplying and dividing by $p(\mathbf{y}|\mathbf{x})$, the true probability of occurance,

$$
= \mathrm{KL}(q(\omega) \,||\, p(\omega)) - \int \Big( \int q(\omega) \big( \log \frac{p(\mathbf{y}|\mathbf{x},\omega)}{p(\mathbf{y}|\mathbf{x})} + \log p(\mathbf{y}|\mathbf{x}) \big) d\omega \Big) d(\mathbf{x},\mathbf{y}).
\tag{S2}
$$

Using $\int q(\omega)\, d\omega = 1$,

$$
\begin{aligned}
&= \mathrm{KL}(q(\omega) \,||\, p(\omega)) - \int \Big( \int q(\omega) \log \frac{p(\mathbf{y}|\mathbf{x},\omega)}{p(\mathbf{y}|\mathbf{x})} d\omega + \log p(\mathbf{y}|\mathbf{x}) \Big) d(\mathbf{x},\mathbf{y}). \\
&= \mathrm{KL}(q(\omega) \,||\, p(\omega)) - \int \int q(\omega) \log \frac{p(\mathbf{y}|\mathbf{x},\omega)}{p(\mathbf{y}|\mathbf{x})} d\omega \; d(\mathbf{x},\mathbf{y}) - \int \log p(\mathbf{y}|\mathbf{x}) d(\mathbf{x},\mathbf{y}).
\end{aligned}
\tag{S3}
$$

As $\int \log p(\mathbf{y}|\mathbf{x}) d(\mathbf{x},\mathbf{y})$ is independent of $\omega$, the variables we are optmizing over, we have,

$$
\propto \mathrm{KL}(q(\omega) \,||\, p(\omega)) - \int \int q(\omega) \log \frac{p(\mathbf{y}|\mathbf{x},\omega)}{p(\mathbf{y}|\mathbf{x})} d\omega \; d(\mathbf{x},\mathbf{y}).
\tag{S4}
$$

APPENDIX B. RESULTS ON SIMPLE MULTI-MODAL 2D DATA.

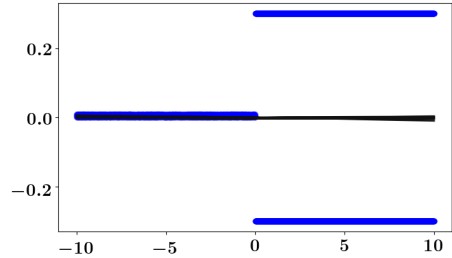 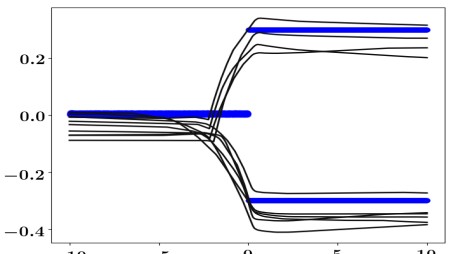

**Figure 7:** Blue: Data points. Black: Sampled models $\hat{\omega} \in q(\omega)$ learned by the Bayes-S approach. All models fit to the mean.

**Figure 8:** Blue: Data points. Black: Sampled models $\hat{\omega} \in q(\omega)$ learned by the Bayes-SL approach. We recover models covering both modes.

We show results on simple multi-modal 2d data as in the motivating example in the introduction. The data consists of two parts: $x \in [-10, 0]$ we have $y = 0$ and $x \in [0, 10]$ we have $y = (-0.3, 0.3)$. The set of models under consideration is a two hidden layer neural network with 256-128 neurons with 50% dropout. We show 10 randomly sampled models from $\hat{\omega} \sim q(\omega)$ learned by the Bayes-S approach in Figure 7 and our Bayes-SL approach in Figure 8 (with $\alpha = 1, \beta = 0$). We assume constant observation uncertainty (=1). We clearly see that our Bayes-SL learns models which cover both modes, while all the models learned by Bayes-S fit to the mean. Clearly showing that our approach can better capture model uncertainty.

APPENDIX C. ADDITIONAL DETAILS AND EVALUATION ON STREET SCENES.

First, we provide additional training details of our Bayes-WD-SL in Table 5.

| Property | Value |
|----------|-------|
| Generator learning rate | $1 \times 10^{-4}$ |
| Discriminator learning rate | $1 \times 10^{-4}$ |
| # Generator updates per iteration | 1 |
| # Discriminator updates per iteration | 1 |

**Table 5:** Additional training details of our Bayes-WD-SL model.

| | Timestep | |
|--------|----------|----------|
| Method | +0.18sec | +0.54sec |
| Last Input (Dialation 10) | 49.4 | 36.9 |
| Luc et al. (2017) (ft) | 59.4 | 47.8 |
| Bayes-WD-SL (ft, mean) | 60.1 | 48.7 |
| Bayes-WD-SL (ft, Top 5%) | **61.6** | **51.3** |

**Table 6:** Additional Comparison to Luc et al. (2017) using the same Dialation 10 approach to generate training segmentations. Note: Fine Tuned (ft) means both approaches are trained to predict at intervals of three frames (0.18 seconds).

Second, we provide additional evaluation on street scenes. In Section 4.2 (Table 1) we use a PSPNet to generate training segmentations for our Bayes-WD-SL model to ensure fair comparison with the state-of-the-art (Seyed et al., 2018). However, the method of Luc et al. (2017) uses a weaker Dialation 10 approach to generate training segmentations. Note that our Bayes-WD-SL model already obtains higher gains in comparison to Luc et al. (2017) with respect the Last Input Baseline, e.g. at +0.54sec, 47.8 - 36.9 = 10.9 mIoU translating to 29.5% gain over the Last Input Baseline of Luc et al. (2017) versus 51.2 - 38.3 = 12.9 mIoU translating to 33.6% gain over the Last Input Baseline of our Bayes-WD-SL model in Table 1. But for fairness, here we additionally include results in Table 6 using the same Dialation 10 approach to generate training segmentations.

We observe that our Bayes-WD-SL model beats the model of Luc et al. (2017) in both short-term (+0.18 sec) and long-term predictions (+0.54 sec). Furthermore, we see that the mean of the Top 5% of the predictions of Bayes-WD-SL leads to much improved results over mean predictions. This again confirms the ability of our Bayes-WD-SL model to capture uncertainty and deal with multi-modal futures.

## APPENDIX D. RESULTS ON HKO PRECIPITATION FORECASTING DATA.

The HKO radar echo dataset consists of weather radar intensity images. We use the train/test split used in Xingjian et al. (2015); Bhattacharyya et al. (2018b). Each sequence consists of 20 frames. We use 5 frames as input and 15 for prediction. Each frame is recorded at an interval of 6 minutes. Therefore, they display considerable uncertainty. We use the same network architecture as used for street scene segmentation Bayes-WD-SL (Figure 2 and with $\alpha = 5, \beta = 1$), but with half the convolutional filters at each level. We compare to the following baselines: 1. A deterministic model (ResG-Mean), 2. A Bayesian model with weight dropout. We report the (oracle) Top-10% scores (best 1 of 10), over the following metrics (Xingjian et al., 2015; Bhattacharyya et al., 2018b), 1. Rainfall-MSE: Rainfall mean squared error, 2. CSI: Critical success index, 3. FAR: False alarm rate, 4. POD: Probability of detection, and 5. Correlation, in Table 7,

Note, that Xingjian et al. (2015); Bhattacharyya et al. (2018b) reports only scores over mean of all samples. Our ResG-Mean model outperforms these state of the art methods, showing the versatility of our model architecture. Our Bayes-WD-SL can outperform the strong ResG-Mean baseline again showing that it learns to capture uncertainty (see Figure 10). In comparison, the Bayes-WD baseline struggles to outperform the ResG-Mean baseline.

We further compare the calibration our Bayes-SL model to the ResG-Mean model in Figure 9. We plot the predicted intensity to the true mean observed intensity. The difference to ResG-Mean model

| Method | Rainfall-MSE | CSI | FAR | POD | Correlation |
|---|---|---|---|---|---|
| Xingjian et al. (2015) (mean) | 1.420 | 0.577 | 0.195 | 0.660 | 0.908 |
| Bhattacharyya et al. (2018b) (mean) | 1.163 | 0.670 | 0.163 | 0.734 | 0.918 |
| ResG-Mean | 1.286 | 0.720 | 0.104 | **0.780** | 0.942 |
| Bayes-WD (Top-10%) | 1.067 | 0.718 | 0.113 | 0.771 | 0.944 |
| Bayes-WD-SL (Top-10%) | **1.033** | **0.721** | **0.102** | **0.780** | **0.945** |

**Table 7:** Evaluation on HKO radar image sequences.

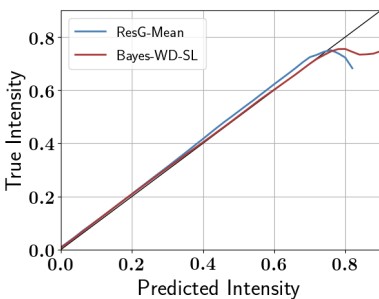

**Figure 9:** Predicted radar intensity to the true mean observed intensity.

is stark in the high intensity region. The RegG-Mean model deviates strongly from the diagonal in this region – it overestimates the radar intensity. In comparison, we see that our Bayes-WD-SL approach stays closer to the diagonal. These results again show that our synthetic likelihood based approach leads to more accurate predictions while not compromising on calibration.

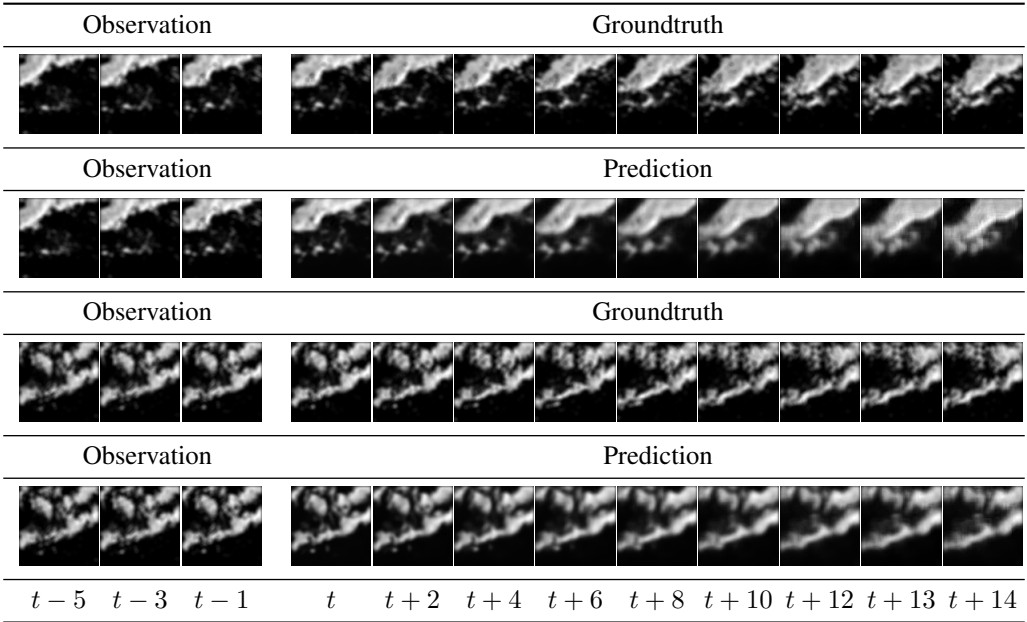

**Figure 10:** Example Top-1 predictions by our Bayes-WD-SL model.

## APPENDIX E. ADDITIONAL ARCHITECTURE DETAILS.

Here, we provide layer-wise details of our generative and discriminative models in Table 8 and Table 9. We provide layer-wise details of the recognition network of the CVAE baseline used in Table 4 (in the main paper). Finally, in Table 11 we show the difference in the number of possible

models using our weight based variational distribution 4 (weight dropout) versus the patch based variational distribution (patch dropout) proposed in Gal & Ghahramani (2016a). The number of patches is calculated using the formula,

$$\text{Input Resolution} \times \text{\# Output Convolutional Filters},$$

because we use convolutional stride 1, padding to ensure same output resolution and each patch is dropped out (in Gal & Ghahramani (2016a)) independently for each convolutional filter. The number of weight parameters is given by the formula,

$$\text{Filter size} \times \text{\# Input Convolutional Filters} \times \text{\# Output Convolutional Filters} + \text{\# Bias}.$$

Table 11 shows that our weight dropout scheme results in significantly lower number of parameters compared to patch dropout Gal & Ghahramani (2016a).

**Details of our generative model.** We show the layer wise details in Table 8.

| Layer | Type | Size | Activation | Input | Output |
|---|---|---|---|---|---|
| $\text{In}_1$ | Input | | | x | $\text{Conv}_{1,1}$ |
| $\text{Conv}_{1,1}$ | Conv2D | 128 | *ReLU* | $\text{In}_1$ | $\text{Conv}_{1,2}$ |
| $\text{Conv}_{1,2}$ | Conv2D | 128 | *ReLU* | $\text{Conv}_{1,1}$ | $\text{Conv}_{1,3}$ |
| $\text{Conv}_{1,3}$ | Conv2D | 128 | *ReLU* | $\text{Conv}_{1,2}$ | $\text{ResConc}_1$ |
| $\text{ResConc}_1$ | Residual Connection | 128 | | $\{\text{Conv}_{1,1}, \text{Conv}_{1,3}\}$ | $\text{MaxPool}_1$ |
| $\text{MaxPool}_1$ | Max Pooling | $2\times2$ | | $\text{ResConc}_1$ | $\text{Conv}_{2,1}$ |
| $\text{Conv}_{2,1}$ | Conv2D | 256 | *ReLU* | $\text{MaxPool}_1$ | $\text{Conv}_{2,2}$ |
| $\text{Conv}_{2,2}$ | Conv2D | 256 | *ReLU* | $\text{Conv}_{2,1}$ | $\text{Conv}_{2,3}$ |
| $\text{Conv}_{2,3}$ | Conv2D | 256 | *ReLU* | $\text{Conv}_{2,2}$ | $\text{ResConc}_2$ |
| $\text{ResConc}_2$ | Residual Connection | 128 | | $\{\text{Conv}_{1,3}, \text{Conv}_{2,3}\}$ | $\text{MaxPool}_2$ |
| $\text{MaxPool}_2$ | Max Pooling | $2\times2$ | | $\text{ResConc}_1$ | $\text{Conv}_{3,1}$ |
| $\text{Conv}_{3,1}$ | Conv2D | 512 | *ReLU* | $\text{MaxPool}_2$ | $\text{Conv}_{3,2}$ |
| $\text{Conv}_{3,2}$ | Conv2D | 512 | *ReLU* | $\text{Conv}_{3,1}$ | $\text{Conv}_{3,3}$ |
| $\text{Conv}_{3,3}$ | Conv2D | 512 | *ReLU* | $\text{Conv}_{3,2}$ | $\text{ResConc}_3$ |
| $\text{ResConc}_3$ | Residual Connection | 128 | | $\{\text{Conv}_{2,3}, \text{Conv}_{3,3}\}$ | $\text{UpSamp}_1$ |
| $\text{UpSamp}_1$ | Up Sampling | $2\times2$ | | $\text{ResConc}_3$ | $\text{Conv}_{4,1}$ |
| $\text{Conv}_{4,1}$ | Conv2D | 256 | *ReLU* | $\text{UpSamp}_1$ | $\text{Conv}_{4,2}$ |
| $\text{Conv}_{4,2}$ | Conv2D | 256 | *ReLU* | $\text{Conv}_{4,1}$ | $\text{Conv}_{4,3}$ |
| $\text{Conv}_{4,3}$ | Conv2D | 256 | *ReLU* | $\text{Conv}_{4,2}$ | $\text{ResConc}_4$ |
| $\text{ResConc}_4$ | Residual Connection | 128 | | $\{\text{Conv}_{3,3}, \text{Conv}_{4,3}\}$ | $\text{UpSamp}_2$ |
| $\text{UpSamp}_2$ | Up Sampling | $2\times2$ | | $\text{ResConc}_3$ | $\text{Conv}_{5,1}$ |
| $\text{Conv}_{5,1}$ | Conv2D | 128 | *ReLU* | $\text{UpSamp}_2$ | $\text{Conv}_{5,2}$ |
| $\text{Conv}_{5,2}$ | Conv2D | 64 | *ReLU* | $\text{Conv}_{5,1}$ | $\text{Conv}_{5,3}$ |
| $\text{Conv}_{5,3}$ | Conv2D | 64 | *ReLU* | $\text{Conv}_{5,2}$ | $\text{Conv}_{5,3}$ |
| $\text{Conv}_6$ | Conv2D | 38 | | $\text{Conv}_{5,3}$ | GaussS |
| GaussS | Gaussian Sampling | | | $\text{Conv}_6$ | y |

**Table 8:** Details our generative model. The final output of $\text{Conv}_6$ is split into mean and variances for sampling as in (6) of the main paper.

**Details of our discriminator model.** We show the layer wise details in Table 9.

**Details of the recognition model used in the CVAE baseline.** We show the layer wise details in Table 10.

| Layer | Type | Size | Activation | Input | Output |
|---|---|---|---|---|---|
| $In_1$ | Input | | | x, y | $Conv_{1,1}$ |
| $Conv_{1,1}$ | Conv2D | 128 | *ReLU* | $In_1$ | $Conv_{1,2}$ |
| $Conv_{1,2}$ | Conv2D | 128 | *ReLU* | $Conv_{1,1}$ | $MaxPool_1$ |
| $MaxPool_1$ | Max Pooling | 2×2 | | $Conv_{1,2}$ | $Conv_{2,1}$ |
| $Conv_{2,1}$ | Conv2D | 256 | *ReLU* | $MaxPool_1$ | $Conv_{2,2}$ |
| $Conv_{2,2}$ | Conv2D | 256 | *ReLU* | $Conv_{2,1}$ | $MaxPool_2$ |
| $MaxPool_2$ | Max Pooling | 2×2 | | $Conv_{2,2}$ | $Conv_{3,1}$ |
| $Conv_{3,1}$ | Conv2D | 512 | *ReLU* | $MaxPool_2$ | $MaxPool_2$ |
| $MaxPool_3$ | Max Pooling | 2×2 | | $Conv_{3,1}$ | $Conv_{4,1}$ |
| $Conv_{4,1}$ | Conv2D | 512 | *ReLU* | $MaxPool_3$ | $MaxPool_4$ |
| $MaxPool_4$ | Max Pooling | 2×2 | | $Conv_{4,1}$ | Flatten |
| Flatten | | | | $MaxPool_4$ | $Dense_1$ |
| $Dense_1$ | Fully Connected | 1024 | *ReLU* | Flatten | $Dense_2$ |
| $Dense_2$ | Fully Connected | 1024 | *ReLU* | $Dense_1$ | Out |
| Out | Fully Connected | - | | $Dense_2$ | |

**Table 9:** Details our discriminator model. The final output Out provides the synthetic likelihoods $\left(\frac{D(x,\hat{y})}{1-D(x,\hat{y})}\right)$.

| Layer | Type | Size | Activation | Input | Output |
|---|---|---|---|---|---|
| $In_1$ | Input | | | x, y | $Conv_{1,1}$ |
| $Conv_{1,1}$ | Conv2D | 128 | *ReLU* | $In_1$ | $Conv_{1,2}$ |
| $Conv_{1,2}$ | Conv2D | 128 | *ReLU* | $Conv_{1,1}$ | $MaxPool_1$ |
| $MaxPool_1$ | Max Pooling | 2×2 | | $Conv_{1,2}$ | $Conv_{2,1}$ |
| $Conv_{2,1}$ | Conv2D | 128 | *ReLU* | $MaxPool_1$ | $Conv_{2,2}$ |
| $Conv_{2,2}$ | Conv2D | 128 | *ReLU* | $Conv_{2,1}$ | $MaxPool_2$ |
| $MaxPool_2$ | Max Pooling | 2×2 | | $Conv_{2,2}$ | $Conv_{3,1}$ |
| $Conv_{3,1}$ | Conv2D | 128 | *ReLU* | $MaxPool_2$ | $Conv_{4,1}$ |
| $Conv_{4,1}$ | Conv2D | 128 | *ReLU* | $Conv_{3,1}$ | $UpSamp_1$ |
| $UpSamp_1$ | Up Sampling | 2×2 | | $Conv_{4,1}$ | $Conv_{5,1}$ |
| $Conv_{5,1}$ | Conv2D | 128 | *ReLU* | $UpSamp_1$ | $UpSamp_2$ |
| $UpSamp_2$ | Up Sampling | 2×2 | | $Conv_{3,2}$ | $Conv_{4,1}$ |
| $Conv_{6,1}$ | Conv2D | 32 | | $UpSamp_2$ | $z_1$ |
| $Conv_{6,2}$ | Conv2D | 32 | | $UpSamp_2$ | $z_2$ |
| $Conv_{6,3}$ | Conv2D | 32 | | $UpSamp_2$ | $z_3$ |

**Table 10:** Details the recognition model used in the CVAE baseline. The final outputs are the Gaussian Noise tensors $z_1, z_2, z_3$.

| Layer | Type | Filter Size | Kernel Size | Resolution | # Patches | # Weights |
|---|---|---|---|---|---|---|
| $In_1$ | Input | | | | | |
| $Conv_{1,1}$ | Conv2D | 128 | 3×3 | 128×256 | 4,194,364 | 87,680 |
| $Conv_{1,2}$ | Conv2D | 128 | 3×3 | 128×256 | 4,194,364 | 147,584 |
| $Conv_{1,3}$ | Conv2D | 128 | 3×3 | 128×256 | 4,194,364 | 147,584 |
| $ResConc_1$ | Residual Connection | 128 | | | | |
| $MaxPool_1$ | Max Pooling | 2×2 | | | | |
| $Conv_{2,1}$ | Conv2D | 256 | 3×3 | 64×128 | 2,097,152 | 290,168 |
| $Conv_{2,2}$ | Conv2D | 256 | 3×3 | 64×128 | 2,097,152 | 590,080 |
| $Conv_{2,3}$ | Conv2D | 256 | 3×3 | 64×128 | 2,097,152 | 590,080 |
| $ResConc_2$ | Residual Connection | 128 | | | | |
| $MaxPool_2$ | Max Pooling | 2×2 | | | | |
| $Conv_{3,1}$ | Conv2D | 512 | 3×3 | 32×64 | 1,048,576 | 1,180,160 |
| $Conv_{3,2}$ | Conv2D | 512 | 3×3 | 32×64 | 1,048,576 | 2,359,808 |
| $Conv_{3,3}$ | Conv2D | 512 | 3×3 | 32×64 | 1,048,576 | 2,359,808 |
| $ResConc_3$ | Residual Connection | 128 | | | | |
| $UpSamp_1$ | Up Sampling | 2×2 | | | | |
| $Conv_{4,1}$ | Conv2D | 256 | 3×3 | 64×128 | 2,097,152 | 1,180,160 |
| $Conv_{4,2}$ | Conv2D | 256 | 3×3 | 64×128 | 2,097,152 | 590,080 |
| $Conv_{4,3}$ | Conv2D | 256 | 3×3 | 64×128 | 2,097,152 | 590,080 |
| $ResConc_4$ | Residual Connection | 128 | | | | |
| $UpSamp_2$ | Up Sampling | 2×2 | | | | |
| $Conv_{5,1}$ | Conv2D | 128 | 3×3 | 128×256 | 4,194,364 | 295,040 |
| $Conv_{5,2}$ | Conv2D | 64 | 3×3 | 128×256 | 2,097,152 | 73,792 |
| $Conv_{5,3}$ | Conv2D | 64 | 3×3 | 128×256 | 2,097,152 | 36,928 |
| $Conv_6$ | Conv2D | 38 | 3×3 | | | |
| GaussS | Gaussian Sampling | | | | | |

| Total possible models ($2^{\#}$) | |
|---|---|
| Gal & Ghahramani (2016a) | **Ours** |
| 36,700,406 | 11,699,192 |
| 36,700,406 | 11,699,192 |

**Table 11:** The difference in the number of possible models using our Weight dropout scheme versus patch dropout Gal & Ghahramani (2016a) on the architecture for street scene prediction Figure 2

| | Total possible models ($2^{\#}$) | | |
|---|---|---|---|
| Model Architecture | Gal & Ghahramani (2016a) | **Ours** | **Reduction by** |
| Street Scene Prediction Figure 2 | 36,700,406 | 11,699,192 | 68.1% |
| Precipitation Forecasting (Appendix D) | 18,350,203 | 5,849,596 | 68.1% |

**Table 12:** Overview of the variational parameters using our Weight dropout scheme versus patch dropout Gal & Ghahramani (2016a) of both architectures for street scene and precipitation forecasting

