# OpenReview forum: "Bayesian Prediction of Future Street Scenes using Synthetic Likelihoods"
_ICLR.cc/2019/Conference_

### Official Review · AnonReviewer3 · 2018-10-31
**a nice applied work**

**Rating:** 6
**Confidence:** 2

**Review:**

The paper presents an application of Bayesian neural networks in predicting
future street scenes. The inference is done by using variational approximation
to the posterior. Moreover, the authors propose to using a synthetic (approximate)
likelihood and the optimization step in variational approxiation is based on a regularization.
These modifications are claimed by the authors that it yields a better results in practice
(more stable, capture the multi-modal nature). Numerical parts in the paper support
the authors' claims: their method outperforms some other state-of-the-art methods.

The presentation is not too hard to follow.
I think this is a nice applied piece, although I have never worked on this applied side.

Minor comment:
In the second sentence, in Section 3.1, page 3,
$f: x \mapsto y$    NOT $f: x \rightarrow y$.
We use the "\rightarrow" for spaces X,Y not for variables.

---

> ### Author Response · Authors · 2018-11-21
> **Typos Corrected.**
>
> We would like to thank AnonReviewer3 for recognizing that our model leads to better results and can capture the multi-modal nature of future scenes. We have addressed the typos that were pointed out. We would be happy to answer any remaining questions.

---

### Official Review · AnonReviewer2 · 2018-11-01
**Interesting improvement to dropout based Bayesian inference**

**Rating:** 8
**Confidence:** 4

**Review:**

The submission considers a disadvantage of a standard dropout-based Bayesian inference approach, namely the pessimization of model uncertainty by means of maximizing the average likelihood for every data sample. The formulation by Gal & Ghahramani is improved upon two-fold: via simplified modeling of the approximating variational distribution (on kernel/bias instead of on patch level), and by using a discriminator (i.e. classifier) for providing a "synthetic" likelihood estimate. The latter relaxes the assumptions such that not every data sample needs to be explained equally well by the models.
Results are demonstrated on a variety of tasks, most prominently street scene forecasting, but also digit completion and precipitation forecasting. The proposed method improves upon the state of the art, while more strongly capturing multi-modality than previous methods.

To the best of my knowledge, this is the first work w.r.t. future prediction with a principled treatment of uncertainty. I find the contributions significant, well described, and the intuition behind them is conveyed convincingly. The experiments in Section 4 (and appendix) yield convincing results on a range of problems.
Clarity of the submission is overall good; Sections 3.1-3.3 treat the contributions in sufficient detail. Descriptions of both generator and discriminator for street scenes (Section 3.4) are sufficiently clear, although I would like to see a more detailed description of the training process (how many iterations for each, learning rate, etc?) for better reproducability.
In Section 3.4, it is not completely clear to me why the future vehicle odometry is provided as an input, in addition to past odometry and past segmentation confidences. I assume this would not be present in a real-world scenario? I also have to admit that I fail to understand Figure 4; at least I cannot see any truly significant differences, unless I heavily zoom in on screen.

Small notes:
- Is the 'y' on the right side of Equation (5) a typo? (should this be 'x'?)
- The second to last sentence at the bottom of page 6 ("Always the comparison...") suffers from weird grammar

---

> ### Author Response · Authors · 2018-11-21
> **Clarifications Provided.**
>
> We would like to thank AnonReviewer2 for recognising that our method is the first work w.r.t. future prediction with a principled treatment of uncertainty. We are pleased that the results in Section 4 (and the Appendix) are convincing.
>
> We now address the concerns and provide clarifications in detail,
>
> * Detailed description of the training process should be provided for better reproducibility.
> - We completely agree and we have added additional details in Appendix C Table 5. We will make the code available at the time of publication.
>
> * it is not completely clear to me why the future vehicle odometry is provided as an input, in addition to past odometry and past segmentation confidences. I assume this would not be present in a real-world scenario?
> - Conditioning on the future odometry is important as the sequences are in the frame of reference of the vehicle. The future path of the vehicle thus has a impact on the future observed sequence. In real world scenarios, the future odometry can be seen as a candidate planned path of the vehicle. Thus, this would help the vehicle obtain more informative predictions conditioned on candidate paths and help the vehicle decide on the optimal path.
>
> * The differences in Figure 4.
> - Figure 4 depicts the long-term uncertainty calibration of our method and baselines -- how well  the predicted probability of a class 0.6sec into the future corresponds to the observed frequency. Uncertainty calibration of long-term predictions is a very important but difficult to achieve property (as we cannot directly optimize for it). Bayesian inference provides the only reliable approach to obtain calibration uncertainties. In Figure 4, we see that  standard dropout bayesian inference approach of Kendall & Gal (2017) already leads to improved results over the non-Bayesian ResG-Mean and CVAE approaches. The improvements are in line with improvements observed in Kendall & Gal (2017). Our Bayes-WD and Bayes-WD-SL further improves upon the approach of Kendall & Gal (2017). Furthermore, note that the differences are significant in the important regions: [0.0,0.2] and [0.6,0.95]. Other methods underestimate the probability of occurrence of classes in the region [0.0,0.2] (they fail to predict certain classes which potentially occur in special cases) and overestimate in the region [0.6,0.95] (they are overly confident of likely classes).
>
> We have corrected the typos that were pointed out. Finally, we thank AnonReviewer2 for voicing her/his concerns and helping us improve our work.  We would be happy to answer any remaining questions.

---

> > ### Comment · AnonReviewer2 · 2018-12-04
> > **Thanks**
> >
> > Thanks for the rebuttal. I think the added clarifications, also the ones given to R1, improve the paper even further. It clearly remains in the "top 50% of accepted papers" group for me.

---

### Official Review · AnonReviewer1 · 2018-11-06
**Some interesting ideas, clarifications needed**

**Rating:** 6
**Confidence:** 4

**Review:**

The work proposes a Bayesian neural network model that is a hybrid between autoencoders and GANs, although it is not presented like that. Specifically, the paper starts from a Bayesian Neural Network model, as presented in Gal and Ghahramani, 2016 and makes two modifications.

First, it proposes to define one Bernoulli variational distribution per weight kernel, instead  of per patch (in the original work there was one Bernoulli distribution per patch kernel). As the paper claims, this reduces the complexity to be exponential to the number of weights, instead of the number of patches, which leads to a much smaller number of possible models. Also, because of this modification the same variational distributions are shared between locations, being closer to the convolutional nature of the model.

The second modification is the introduction of synthetic likelihoods. Specifically, in the original network the variational distributions are designed such that the KL-divergence of the true posterior p(ω|X, y) and the approximate posterior q(ω) is minimiezd. This leads to the optimizer encouraging the final model to be close to the mean, thus resulting in less diversity. By re-formulating the KL-divergence, the final objective can be written such that it depends on the likelihood ratio between generated/"fake" samples and "true" data samples. This ratio can then be approximated by a GAN-like discriminator. As the optimizer now is forced to care about the ratio instead of individual samples, the model is more diverse.

Both modifications present some interesting ideas. Specifically, the number of variational parameters is reduced, thus the final models could be much better scaleable. Also, using synthetic likelihoods in a Bayesian context is novel, to the best of my knowledge, and does seem to be somewhat empirically justified.

The negative points of the paper are the following.

- The precise novelty of the first modification is not clearly explained. Indeed, the number of possible models with the proposed approach is reduced. However, what is the degree to which it is reduced. With some rough calculations, for an input image of resolution 224x224, with a kernel size of 3x3 and stride 1, there should be about 90x90 patches. That is roughly a complexity of O(N^2) ~ 8K (N is the number of patches). Consider the proposed variational distributions with 512 outputting channels, this amount to 3x3x512 ~ 4.5K. So, is the advantage mostly when the spatial resolution of the image is very high? What about intermediate layers, where the resolution is typically smaller?

- Although seemingly ok, the experimental validation has some unclarities.
  + First, it is not clear whether it is fair in the MNIST experiment to report results only from the best sampled model, especially considering that the difference from the CVAE baseline is only 0.5%. The standard deviation should also be reported.
  + In Table 2 it is not clear what is compared against what. There are three different variants of the proposed model. The WD-SL does exactly on par with the Bayes-Standard (although for some reason the boldface font is used only for the proposed method. The improvement appears to come from the synthetic likelihoods. Then, there is another "fine-tuned" variant for which only a single time step is reported, namely +0.54 sec. Why not report numbers for all three future time steps? Then, the fine-tuned version (WD-SL-ft) is clearly better than the best baselines of Luc et al., however, the segmentation networks are also quite different (about 7% difference in mIoU), so it is not clear if the improvement really comes from the synhetic likelihoods or from the better segmentation network. In short, the only configuration that appears to be convincing as is is for the 0.06 sec. I would ask the authors to fill in the blank X spots and repeat fair experiments with the baseline.

- Generally, although the paper is ok written, there are several unclarities.
  + Z_K in eq. (4) is not defined, although I guess it's the matrix of the z^{i, j}_{k, k'}
  + In eq (6) is the z x σ a matrix or a scalar operation? Is z a matrix or a scalar?
  + The whole section 3.4 is confusing and it feels as if it is there to fill up space. There is a rather intricate architecture, but it is not clear where it is used. In the first experment a simple fully connected network is used. In the second experiment a ResNet is used. So, where the section 3.4 model used?
  + In the first experiment a fully connected network is used, although the first novelty is about convolutions. I suppose the convolutions are not used here? If not, is that a fair experiment to outline the contributions of the method?
  + It is not clear why considering the mean of the best 5% predictions helps with evaluating the predicted uncertainty? I understand that this follows by the citation, but still an explanation is needed.

All in all, there are some interesting ideas, however, clarifications are required before considering acceptance.

---

> ### Author Response · Authors · 2018-11-21
> **Clarifications provided. [1/2]**
>
> We would like to thank AnonReviewer1 for finding our work interesting especially the novel ideas of Weight Dropout and Synthetic Likelihoods in a Bayesian context.
>
> We now address the concerns and provide clarifications in detail,
>
> * The proposed model is a auto-encoder GAN.
> - We would like to clarify that our model is not an auto-encoder GAN. Although,
> the objective function (11) does bear resemblance to the objective typically minimized by auto-encoder GANs. But, we do not use auto-encoders -- our models do not have explicit latent spaces. More importantly, unlike auto-encoder GANs which learn one single model for prediction, we learn the distribution of likely models in a Bayesian framework.
>
> * It is not clear where the architecture of section 3.4 is used.
> - We would like to clarify that this is the main architecture for all the street scene prediction experiments -- used by the ResG-Mean, Bayes-S, Bayes-WD and Bayes-WD-SL models. The architecture of section 3.4 is one and the same as the ResNet architecture. We have clarified this in the text.
>
>
> * Why considering the mean of the best 5% predictions helps with evaluating the predicted uncertainty?
> - As the future becomes increasingly uncertain, we would like to capture all the likely future outcomes rather than just the mean, as mean predictions drift further and further from the groundtruth into the future. As mentioned in the main paper, considering the mean of (oracle) best 5% of predictions helps us evaluate whether the learned model distribution contains likely models corresponding to the groundtruth. As this metric (also used by e.g. Lee et al. (2017) and Bhattacharyya et al. (2018b)) is also averaged across the test set, a higher value shows that the model distribution is able to better anticipate all the varied futures that occur in the test set.
>
> * The precise novelty of the first modification is not clearly explained.
> - The advantage of our proposed Weight Dropout scheme  does depend on the spatial resolution. The gain at a certain layer is significant if the spatial resolution is greater than the product of the number of filters  of in the current layer and the previous layer. We gain ~28x compared to standard (patch) dropout in the first and last group of layer of our model (Figure 2) and ~3x times in the second and fourth group of layers. While we do not gain in the third group of layers -- where the spatial resolution is the lowest, overall we gain ~3x compared to Standard Dropout. Thus, while it is true that the gains are mostly when the spatial resolution is high (typically more than 32x64). But, as the total number of parameters are also the highest when the spatial resolution is high, overall we make significant gains in both the convolutional architectures used for street scene and precipitation forecasting -- we have 68% lower total number of variational parameters compared to the Standard Dropout scheme of Gal & Ghahramani (2016a). We have added Table 11 Appendix E, where we discuss the gains at each layer in detail for the architecture on street scene prediction. Finally, in Table 12 we provide an overview of the variational parameters of both architectures for street scene and precipitation forecasting .
>
> * MNIST experiment reports results only from the best sampled model.
> - Note that we use a classifier to decide if the prediction comes from the correct mode. This is because it is difficult to decide with simple measures like L1/2 distance whether the prediction is even a coherent digit.  We use the best sampled model to maintain the crispness of the predicted images. Taking e.g. the mean of top 10 of 100 models, would lead to the generation of blurry digits potentially unrecognizable by a classifier. Furthermore, this same criteria is applied across all models, therefore, we do not give any unfair advantage to our model. Furthermore, we have updated Figure 3 to report the mean and standard deviation of the Top 10% metric across 10 splits of 1000 examples  from the MNIST test set. The results confirm the advantage of our Bayes-SL model over the Bayes-S and CVAE models.

---

> > ### Author Response · Authors · 2018-11-21
> > **Clarifications provided. [2/2]**
> >
> > * In Table 2 it is not clear what is compared against what.
> > - Here, we compare our proposed model against four important baselines - two state-of-the-art approaches of Luc et al. (2017) and Seyed et al. (2018), the standard dropout approach of Kendall & Gal (2017) (Bayes-S, using the same model architecture as our Bayes-WD-SL - Figure 2) and a baseline that uses Weight Dropout (Bayes-WD) but not synthetic likelihoods. We have updated the Table 2 at the additional time-steps as requested.
> >
> > * The Bayes-WD-SL does exactly on par with the Bayes-Standard.
> > - We have added the additional results for +0.54sec in Table 2. We see that the mean prediction from the Bayes-WD-SL model has an advantage of 2.9 mIoU at 0.06sec and 0.2 mIoU at 0.54secs. The performance advantage of Bayes-WD-SL over Bayes-S(Standard) in this case shows that the ability to better model uncertainty does not come at the cost of lower mean performance. At larger time-steps as the future becomes increasingly uncertain, mean predictions (mean of all likely futures) drift further from the ground-truth. Therefore, we evaluate the models on their (more important) ability to capture the uncertainty of the future in Table 2 (last two rows) and Table 3 (full). We see clearly in Table 3, that Bayes-WD-SL shows a large performance advantage over the Bayes-S(Standard) model in capturing uncertainty with respect to both the Top 5% of predictions and the CLL metric (same criteria as in Lee et al. (2017) and Bhattacharyya et al. (2018b)). This shows that Bayes-WD-SL model can much better capture the range of likely futures.  Similarly, the Bayes-WD also performs better compared to the Bayes-Standard model. This shows the advantage of both our novel components - Weight Dropout and (more significantly) Synthetic Likelihoods in the ability to capture uncertainty.  We have updated the text in Section 4.2 to highlight these points.
> >
> > * Fair comparison to Luc et al. (2017)
> > - In Table 1,  we use a PSPNet to generate training segmentations for our Bayes-WD-SL model to ensure fair comparison with the state-of-the-art Seyed et al. Note, that our Bayes-WD-SL model already obtains higher gains in comparison to Luc et al. with respect the Last Input Baseline, e.g. at +0.54sec, 47.8 - 36.9 = 10.9 mIoU translating to 29.5% gain over the Last Input Baseline of Luc et al. versus 51.2 - 38.3 = 12.9 mIoU translating to 33.6% gain over the Last Input Baseline of our Bayes-WD-SL model in Table 1. But as suggested for fairness,  we additionally include results in Table 6, Appendix C using the same Dialation 10 approach to generate training segmentations. We observe that our Bayes-WD-SL model beats the model of Luc et al. (2017) in both short-term (+0.18 sec) and long-term predictions (+0.54 sec). Furthermore, we see that the mean of the Top 5% of the predictions of Bayes-WD-SL leads to much improved results over mean predictions. This again confirms the ability of our Bayes-WD-SL model to capture uncertainty and deal with multi-modal futures.
> >
> > * Unclarities.
> > - We have defined Z_K in eq. (4) in the text.
> > In eq (6) is the z x σ a matrix or a scalar operation?
> > - It is a matrix operation as z is vector. We have clarified this in the text.
> > Convolutions are not used in the first experiment?
> > - We do not use convolutions in the first experiment as we want to highlight the advantage of synthetic likelihoods in modelling uncertainty. The results of the experiment show that synthetic likelihoods are applicable across model types and can successfully model uncertainty. Note that we use weight dropout (our first novelty) in the main experiments on street scenes.
> >
> > Finally, we thank AnonReviewer1 for voicing her/his concerns and helping us improve our work.  We would be happy to answer any remaining questions.

---

> > > ### Comment · AnonReviewer1 · 2018-12-04
> > > **Nice rebuttal**
> > >
> > > I would first like to congratulate the reviewers for the ir rebuttal and the clarifications. I do believe that the clarifications and changes made the paper better. My only problem is with using the top 5% predictions to capture the uncertainty. I understand the point of using the top 5% predictions, but I am still not entirely convinced that this measures the uncertainty, in that uncertainty is the capacity of the model to predict the various future outcomes. Evaluating the top 5%, however, measures if in the "longer" future there are some predicted trajectories that are still close to the ground truth, which is clearly a very different concept. That said, I do not have a better proposal, so, for now, I rest my case.

---

> > > > ### Author Response · Authors · 2018-12-06
> > > > **Thank you and Further Clarifications about the Top 5% Criterion.**
> > > >
> > > > Thank you for your response. We are glad that our clarifications and changes have made the paper better. Regarding the Oracle Top 5% criterion: It is true that for a single test sequence, the Oracle Top 5% criterion measures if in the "longer" future there are some predicted trajectories that are still close to the ground truth. However, note that it is the Top 5% of a limited “budget” of 100 predictions and this criterion is averaged over the test-set. Therefore, to obtain a low Top 5% error, our model must generate likely (due to limited budget) future trajectories for all test-set examples (because it is averaged across the test-set) that are close to the ground-truth. In other words, the model must be able to predict the likely future outcomes. A model which assigns low probability to likely futures will not be able to generate future trajectories corresponding to the ground truth in the limited budget of 100 samples per test examples for the vast majority of the test-set sequences -- leading to a high Top 5% error. Therefore, the Top 5% error can measure the uncertainty. It has also been used in prior work for a relative comparison of techniques (Lee et al. (2017) and Bhattacharyya et al. (2018b)). We will better motivate the Oracle Top 5% criterion in the final version. Furthermore, note that we also include the Conditional log-likelihood (CLL) metric in Table 3 as an additional metric to measure uncertainty.

---

### Author Response · Authors · 2018-11-29
**Overview of Updates in the Revision.**

- In Section 3.4 of the main paper,  we clarify that the ResNet based architecture in Figure 2 is the architecture used for street scene prediction.

- In Appendix E, Tables 11 and 12, we clearly illustrate the reduction in variational parameters enabled by our novel Weight Dropout scheme.

- Figure 3 in the main paper has been updated to include both the mean accuracy and the standard deviation across 10 splits of 1000 MNIST test set examples -- confirming the advantage of our proposed synthetic likelihoods.

- Table 1 (and corresponding text) in the main paper has been updated to include results at additional time-steps. These results and the results in Table 3, confirm that our model outperforms state-of-the-art approaches.

- In Appendix C, Table 6, we additionally compare our approach to that of Luc et. al. (2017) using the same Dilation 10 method to generate training sequences.

- In Appendix C, Table 5, additional training details have been added.

- Finally, all typos that were pointed out have been corrected.

---

### Author Response · Authors · 2019-05-28
**Code and Data**

The code and data is available here: https://github.com/apratimbhattacharyya18/seg_pred

---

### Meta-Review · Area_Chair1 · 2018-12-13
**A good paper to improve diversity of existing Bayesian deep learning methods**

**Confidence:** 5
**Recommendation:** Accept (Poster)

**Metareview:**

This paper proposes a method to encourage diversity of Bayesian dropout method. A discriminator is used to facilitate diversity, which the method deal with multi-modality. Empirical results show good improvement over existing methods. This is a good paper and should be accepted.